# Using Team-Based Learning to Teach Pharmacology within the Medical Curriculum

**DOI:** 10.3390/pharmacy12030091

**Published:** 2024-06-12

**Authors:** Nora L. D. Luitjes, Gisela J. van der Velden, Rahul Pandit

**Affiliations:** 1Department of Translational Neuroscience, Brain Center Rudolf Magnus, University Medical Center Utrecht, Utrecht University, 3584 CG Utrecht, The Netherlands; n.l.d.luitjes@students.uu.nl; 2Education Center, University Medical Center Utrecht, Utrecht University, 3584 CX Utrecht, The Netherlands; g.j.vandervelden-2@umcutrecht.nl

**Keywords:** team-based learning, active learning, pharmacology education, student performance, learning effectiveness

## Abstract

The primary goal of pharmacology teaching is to prepare medical students to prescribe medications both safely and efficiently. At the Utrecht University Medical School, pharmacology is integrated into the three-year bachelor’s curriculum, primarily through large group sessions with limited interaction. A recent evaluation highlighted students’ appreciation for pharmacology teaching, but students admitted to attending these teaching moments unprepared, resulting in passive learning. To address this, team-based learning (TBL) was implemented to facilitate learning through interaction, critical thinking, problem solving and reflection through six steps, from superficial to deeper cognitive learning. This study, conducted over two academic years, assessed students’ perception and performance regarding TBL. Analysis of a digital questionnaire using a 5-point Likert scale showed high student satisfaction with TBL as a teaching methodology. However, confidence in pharmacology knowledge following TBL was moderate. TBL attendees outperformed non-attendees in pharmacology-related exam questions, indicating that TBL has a positive impact on student performance. We conclude that TBL is an engaging and effective method for pharmacology education, positively influencing student learning and performance. This method could be broadly applied for teaching pharmacology within the medical curriculum or other biomedical programs.

## 1. Introduction

The primary goal of pharmacology teaching within the undergraduate medical curriculum is to prepare medical students to prescribe medications both safely and efficiently. Failure to achieve this goal can lead to medication errors, thereby increasing patient morbidity and mortality rates [1]. It is in fact known that inadequacies in the undergraduate curriculum can lead to medication errors [2]. Students are known to struggle with pharmacology, as the subject requires integration of knowledge from various other disciplines such as physiology and pathophysiology [3]. As every patient requires a different approach, understanding the deeper concepts of pharmacology is essential for addressing clinical problems, especially when there are no protocols for specific situations.

To ensure optimal training of medical students, the Utrecht University Medical School in The Netherlands has integrated pharmacology learning throughout various thematic courses within the three-year bachelor’s curriculum (pharmacology longitudinal thread). In these courses, Pharmacology is taught together with other subjects such as physiology, anatomy, biochemistry and pathology [4]. The majority of the teaching hours devoted to pharmacology are, however, taught in large groups (>100 students) with limited teacher–student and student–student interaction. In a recent evaluation of the curriculum, students expressed their appreciation for pharmacology education, but students admit to attending these classes unprepared, resulting in passive learning [4]. Although lectures can be an excellent way to explain difficult concepts, they can also give a false notion of grasping the subject matter [5]. This can be potentially harmful within the medical context. Although prescribing guidelines are accessible and commonly used by physicians, such guidelines do not necessarily include all clinical scenarios, thereby requiring physicians to possess the knowledge and skills to make exceptions or compare diverse clinical protocols when prescribing medications. Translating this to medical education, it is important that students gain a deeper understanding of pharmacological concepts and can apply them in a variety of situations.

Interestingly, even though students following active learning may feel they learn less, their performance is often better [5]. The replacement of passive acceptance and repetition of knowledge with active engagement, not only with peers but also with the subject matter itself, improves students’ performance in exams [6,7].

### 1.1. The Concept and Biology of Active Learning

According to Kyriacou and Marshall (1989), active learning is “a quality of the pupils’ mental experience in which there is active intellectual involvement in the learning experience characterized by increased insight” [6]. The process of memorizing learning material relies on the learning, consolidation, and reconsolidation of knowledge [8]. When new knowledge is introduced, the protein synthesis involved in the biochemical storage of memory is rather unstable. During the process of memory consolidation, a memory converts into a more stable and resilient form and will transfer from the hippocampus to the neocortex [9]. Additionally, memory reconsolidation involves actively recalling previously consolidated knowledge, resulting in highly stable and easily retrievable information [10]. To achieve successful learning outcomes, students need to actively engage in both cognitive and metacognitive processes [8].

### 1.2. Team-Based Learning as a Form of Active Learning in Medical Education

There are numerous pedagogical strategies to provoke active learning, of which team-based learning (TBL) is one [11,12]. TBL was developed by Professor Michaelson in 1979 as a means to improve teaching for larger class sizes [11]. Since then, TBL has been used worldwide in multiple disciplines, mostly in health sciences [8,11], and it contributes to critical thinking and effective collaboration [13,14,15]. TBL provides students with individual work, collaborative teamwork, and immediate feedback to actively assess their conceptual understanding. These activities are conducted by students in small groups, with one instructor with expertise in the content overseeing the entire class [11]. Learning through TBL is an active process and based on the reconstruction of knowledge. To experience knowledge, students must actively apply the knowledge in an authentic problem [16,17]. TBL is based on six steps from superficial cognitive learning to deeper cognitive learning, following the six categories of Bloom’s taxonomy: knowledge, comprehension, application, analysis, synthesis and evaluation [18]. Briefly, in the first step of TBL, students complete a preparatory assignment. This step is followed by the individual readiness assurance test (iRAT), which consists of a set of multiple-choice questions to test individual knowledge. Following this, students answer the same set of questions but as a team, via the team-readiness assurance test (tRAT). In the fourth step, the instructor clarifies any topics students find challenging. In the fifth step, teams solve an authentic clinical problem as a group. Finally, in the last step, students reflect and evaluate their learning experience [11].

The TBL-approach is based on the constructivist learning theory, which emphasizes the importance of experiential learning to achieve long-term integration of new knowledge [13]. The constructivist learning theory is based on four principles. First, the student’s learning is the primary focus. To optimize this process, the learning environment should provide the student with adequate challenges to stimulate the student’s knowledge uptake [17]. Second, students are encouraged to actively interact with the study material through problem solving cases. As a result, students can apply their knowledge to get a deeper understanding of the subject matter as it encourages the reconsolidation of knowledge, which is essential for long-term retention [8]. Third, teamwork is an essential component of the TBL-approach as this collaborative learning stimulates new experiences with knowledge, leading to new insights. Lastly, reflection plays a crucial role in the TBL approach. This reflection includes both strengths and weaknesses of their learning experiences which can be applied or improved in future learning processes [13,16].

There are many elements in TBL which could contribute to its success, but there is no unanimous consensus in the elements of TBL that are considered crucial. Some authors underpin the importance of carefully formed and managed teams, frequent and timely feedback, problem solving, and student peer evaluation [12]. Others have identified competitiveness between the groups as an important factor [19]. However, these recommendations are based on the experiences of teachers, and explicit research has not been conducted to understand the individual importance of the various components of TBL. The educational literature, however, shows that several elements of TBL, such as formative assessment, immediate feedback and teamwork, could be contributing factors [20].

### 1.3. Understanding Team-Based Learning in Our Context

Based on literature, TBL could be a valuable addition to the teaching of pharmacology within our medical curriculum. As teaching is contextual [21], educational practices successfully implemented at one institution do not necessarily have to be equally successful in other institutions. Institutional differences in the use of TBL can be attributed to social and cultural influences, and student openness to teaching methods [22]. Changing the entire pharmacology curriculum to incorporate TBL is time-consuming and costly. Therefore, it is important to first understand the strengths, weaknesses and effectiveness of TBL within our educational context.

The goal of the current study is to understand the effectiveness of TBL as an educational tool for teaching pharmacology to first-year medical students at Utrecht University Medical School. In the current study, a modified TBL approach was used to teach cardiovascular pharmacology, where the six stages of TBL were applied. First, we investigated how students perceived the various components of TBL as a teaching/learning methodology. As preference for a teaching method does not have to necessarily translate to improved performance [5], as a second goal, we investigated the effects of TBL on student performance in the final exam of the course.

## 2. Materials and Methods

### 2.1. Study Setting

The study was conducted during the academic years 2021–2022 and 2022–2023 at Utrecht University Medical School. The medical program at Utrecht University is structured into a three-year bachelor’s and a subsequent three-year master’s degree program. TBL was implemented within a first-year bachelor’s course focused on cardiovascular diseases. The course is offered at the end of the academic year.

The course on cardiovascular diseases comprised five weeks, during which students were exposed to various groups of cardiovascular diseases, their characteristics and their treatment. Pharmacological aspects within this course were divided into lectures (no prior preparation was required), self-study (independent preparation with books and guiding questions), and a TBL session (see Section 2.3. Procedure for more details). Figure 1 provides a detailed structure of the course.

For all students, passing the course on cardiovascular diseases is obligatory to proceed to the second year of the bachelor’s degree in medicine. The course consists of plenary lectures (±300 students), interactive lectures (48–150 students), practical studies, and small-group teaching (25 students). While attendance for all lectures, whether plenary or interactive, is optional, students are required to participate in at least 80% of the small-group teaching sessions. TBL was originally designed for large groups [11]; however, in our experience, larger group sizes result in lower levels of interaction between the students and between the instructor and students. To ensure sufficient interaction during the TBL session, a group size with a maximum of 48 students was chosen. A single instructor (2021–2022) and two instructors (2022–2023) were involved in teaching all TBL sessions on cardiovascular pharmacology. The TBL session was offered in the format of an interactive lecture and was likewise considered optional for attendance.

### 2.2. Study Population

During the academic years of 2021–2022 (*n* = 348) and 2022–2023 (*n* = 338), a total of 686 medical students were enrolled in the first-year course on cardiovascular diseases at Utrecht University Medical School and subsequently completed the final exam at the end of week 5 (Figure 1). As all students had the possibility to follow the TBL session, two distinct groups were formed for data collection: TBL attendees (*n* = 196) and non-TBL attendees (*n* = 490). In 2021–2022, 84 students (31.8% of the cohort) opted to follow TBL education, and in 2022–2023 there were 112 students (33.1% of the cohort). The TBL session was non-obligatory (see Section 2.4., Statistical Analysis, for more details), and all students received the same number of credits if they met the attendance criteria for the course and passed the final exam.

### 2.3. Procedure

Within the framework of the course, the TBL session comprised a structured sequence of six steps followed by step seven, the final exam [11,12], (Figure 2). Educational tips published previously were used to develop the teaching materials. In Section 4.1. Limitations, the strengths and weaknesses of the study approach are discussed in detail. Except for step 1 of TBL (participation) and step 7 (final exam), all other steps of the study procedure were completed during the TBL session lasting 1 h 45 min.

The TBL steps were as follows:Preparation: Two preparatory lectures delved into the pharmacological aspects of antianginal, antithrombotic, and antiarrhythmic drugs. Additionally, as preparation for TBL following these lectures, a preparatory study assignment was provided on heart failure medications. All educational activities were optional, with attendance prior to the TBL session left unrecorded. The preparatory study assignments were accessible for all students and included readings. The non-TBL group also had access to the same reading materials, and the slides of the TBL session were shared following the classes.Individual readiness assurance test (iRAT): PScribe, an electronic prescribing system was used for the iRAT. The iRAT consisted of seven questions covering all pharmacological topics of the course. Individual scores were recorded, which additionally served to note participation (attendance). The iRAT questions were closed-ended and were a mix of single and multiple correct answers for 2021–2022 and single correct answers for 2022–2023. For questions with multiple correct answers, a part of the full score was awarded for partially correct answers. Questions were at the level of understanding, applying or analyzing, according to Bloom’s taxonomy.Team readiness assurance test (tRAT): For the tRAT, students in groups discussed the same questions as during the iRAT and had to come to an agreement on the correct answers to the questions. This step differed from the traditional tRAT, as teams were not predefined, the tRAT scores were not formally registered, and the element of competition was not used in our setting. The reasons for the latter are two: we wanted to favor a low threshold for initiating discussion and engagement, and we also wanted to minimize administrative burden.Instructor clarification: topics considered complex by students were discussed with the entire class and clarified by the instructor.Team application: Students engaged in a collaborative problem-solving exercise involving a clinical case. For this, students were shown a patient video, followed by a set of questions, which students had to answer in groups.Feedback questionnaire: With the help of the online audience response system Wooclap, feedback on the TBL session on cardiovascular pharmacology was gathered through an anonymous digital questionnaire immediately at the end of the TBL session. The questionnaire comprised eight statements where students had to express their level of (dis)agreement on a 5-point Likert scale (1 = strongly disagree, 2 = disagree, 3 = neutral, 4 = agree and 5 = strongly agree). These statements were divided into two categories: (1) students’ own perception on knowledge and (2) feedback on using TBL as a learning method. Participation in the questionnaire via Wooclap was voluntary. There was no incentive provided to participate.Final exam: All students (TBL attendees and non-TBL attendees) took the final exam for the course on cardiovascular diseases via the digital examination platform Testvision. In the final exam, the content on cardiovascular pharmacology was assessed through seven questions in 2021–2022 and eight questions in 2022–2023. Closed-ended questions with single or multiple correct answers were used for this purpose. Similar to the iRAT, for questions with multiple correct answers, partially correct answers were rewarded with a part of the full score. Additionally, here, questions were at the level of understanding, applying or analyzing, according to Bloom’s taxonomy.

### 2.4. Statistical Analysis

Based on the research objectives, different approaches were undertaken to analyze the data. To understand students’ perception of TBL as a learning method, data from the online Wooclap questionnaire were analyzed. The 5-point Likert scale data were used as ordinal variables, and the mean and median were determined. To evaluate the efficacy of TBL as a teaching methodology, three sets of pre-defined comparisons were performed. Initially, the Kolmogorov–Smirnov test was applied to check for normality of distribution, showing that the numerical variables exhibited differing distribution patterns. The statistical approach per research question was as follows:The impact of TBL on students’ understanding of the subject matter was investigated by comparing final exam scores between TBL attendees and non-attendees, using the Mann–Whitney U test. Median scores are reported in percentages of correct answers.The TBL attendees’ individual performance in the final exam was examined by comparing their iRAT scores and final exam scores using the Mann–Whitney U test. Median scores are reported in percentages of correct answers.The correlation between iRAT scores and exam scores, to understand whether the performance in the former correlated with performance in the latter, was explored using Spearman’s correlation coefficient.The difficulty index of the individual questions (*p*-value) of both the iRAT and the final examination was analyzed to understand whether a possible disparity in the level of difficulty could explain the differences in student performance. The *p*-value for questions for one correct answer indicates the proportion of students who correctly answered the question. For questions with multiple possible answers or open-ended questions, the *p*-value indicates the proportion of the points obtained by students for a particular question [23]. In both cases, *p*-values range from 0 to 1, with easier items resulting in higher *p*-values. Differences between the *p*-values (iRAT vs. final exam) were tested using the Mann–Whitney U test.

The threshold for statistical significance was determined as a *p*-value lower than 0.05. Statistical analyses were conducted using IBM SPSS Statistics Version 29.0, and graphical representations were generated using GraphPad Prism Version 10.1.1.

### 2.5. Ethical Approval

Ethical approval for this study was obtained from the Dutch Association for Medical Education Ethical Review Board (NVMO: 2022.6.2). Permission was provided to use iRAT scores from the online platform Pscribe and individual student scores only on pharmacology questions of the final exam and to anonymously obtain feedback on the TBL session. For this reason, no data were collected on the student characteristics (e.g., age, number of times repeating the course) as this would infringe upon the privacy of the students, and access to this sensitive information was not needed for the goal of the study. Data that were non-anonymous were first coded and subsequently analyzed.

## 3. Results

### 3.1. Student Perception Regarding Team-Based Learning

In 2021–2022 and 2022–2023, a total of 196 students who attended the TBL sessions were invited to participate in the questionnaire, of whom 143 (73%) filled it in. All participants completed the entire questionnaire. The Cronbach’s alpha value for the entire questionnaire confirmed that the questionnaire reached high reliability (8 items; Cronbach α = 0.821).

The findings of the questionnaire are summarized in Table 1. In general, students highly valued both the organization and the content of the TBL session. Additionally, students were highly positive about two essential elements of TBL: collaborating with peers to solve problems and using formative assessment. Similarly, the use of a patient video as a part of the clinical case study was appreciated by students. Interestingly, although students had the feeling that the TBL session helped them in achieving the learning goals, students did not feel confident about their knowledge. The statement ‘I think I have sufficient knowledge on this subject.’, scored the lowest, with a mean of 3.2 on the 5-point Likert scale.

### 3.2. Student Performance with Team-Based Learning versus No Team-Based Learning

In 2021–2022 and 2022–2023, the average median score on the pharmacology-related final exam questions for TBL attendees (median (IQR) = 77.14 (67.64–83.29) and 78.22 (65.39–88.22), respectively) was higher compared to non-attendees (median (IQR) = 62.43 (50.54–73.79) and 66.61 (54.50–79.19), respectively). These differences were statistically significant (U = 6452.50, *p* < 0.001 for 2021–2022; U = 8445.00, *p* < 0.001 for 2022–2023). TBL attendees had significantly higher mean ranks (2021–2022: 229.68; 2022–2023: 207.10) compared to non-attendees (2021–2022: 156.94; 2022–2023: 150.87) (Figure 3a).

### 3.3. The iRAT Scores versus the Final Exam Scores

In 2021–2022 and 2022–2023, the median score on the pharmacology-related final exam questions (median (IQR) = 77.14 (67.64–83.29) and 78.22 (65.39–88.22), respectively) was higher compared to the average iRAT score (median (IQR) = 40.25 (24.05–54.73) and 42.86 (28.57–57.14), respectively). Significant improvement in exam performance over the course was evident among the TBL attendees (U = 696.00; *p* < 0.001 for 2021–2022; U = 2024.00; *p* < 0.001 for 2022–2023). The pharmacology-related final exam questions had significantly higher mean ranks (2021–2022: 118.21; 2022–2023: 150.43) compared to the iRAT scores (2021–2022: 50.79; 2022–2023: 74.57) (Figure 3b).

### 3.4. The Relationship between Individual Results during the iRAT and the Final Exam

A Spearman’s rank-order correlation was performed to determine the relationship between individual results during the iRAT and the final exam. A disparity in performance between TBL attendees and non-TBL attendees underscores the significance of TBL in facilitating knowledge application and problem-solving skills essential for success in pharmacology examinations. For both academic years, the correlations were found to not be significant (2021–2022 (rs(82) = 0.089, *p* = 0.423) and 2022–2023 (rs(110) = 0.005, *p* = 0.956).

### 3.5. The Difficulty Index (p-Value) of the iRAT and Final Examination Questions

In 2021–2022 and 2022–2023, the average *p*-values of the iRAT (median (IQR) = 0.42 (0.24–0.53) and 0.43 (0.30–0.55), respectively) were lower compared to the final examination (median (IQR) = 0.63 (0.58–0.77) and 0.70 (0.49–0.74), respectively). These differences were statistically significant (U = 7.00, *p* = 0.025 for 2021–2022; U = 10.00, *p* = 0.021 for 2022–2023). The iRAT questions had significantly lower mean ranks (2021–2022: 5.00; 2022–2023: 5.75) compared to the final examination questions (2021–2022: 10.00; 2022–2023: 11.25).

## 4. Discussion

Student perceptions of TBL indicate overall appreciation for TBL as a teaching methodology. Students appreciate the structured format, the organizational quality, and the informative content of the TBL session. Furthermore, students express that the TBL session aided in achieving the learning objectives and contributed to their pharmacological knowledge. However, it is noteworthy that students express only moderate confidence in their pharmacological knowledge after participating in the TBL session, as indicated by the mean score of 3.2 on the 5-point Likert scale. This observation is not surprising, given the complexity of clinical problems discussed during the TBL session, which might give students the perception that their knowledge is insufficient. This implies that although students may find TBL engaging and beneficial, additional efforts might be necessary to increase their confidence in independently applying pharmacological concepts. In future, score assessments and open-ended questions could enrich the feedback received, although increasing the length of the questionnaire could lead to lower responses [23].

Interestingly, despite the moderate confidence levels, TBL had a significantly positive impact on student performance on the pharmacology-related final exam questions. TBL attendees outperformed non-attendees, with notable differences in final exam scores. This underscores the efficacy of TBL in enhancing students’ comprehension and retention of pharmacological concepts. Moreover, the study revealed a substantial academic advantage associated with participation in TBL. Not only did students who followed TBL perform better in the final exam compared to students who did not follow TBL, the individual performance within the TBL group also improved. The mean final exam scores on the pharmacology-related questions were significantly higher than the mean iRAT scores. These results are consistent across both academic years 2021–2022 and 2022–2023, indicating a consistent effect of TBL and suggesting that TBL increases students’ uptake of pharmacology knowledge. Interestingly, although the difficulty index of the questions of the final exam was lower compared to the iRAT, non-TBL attendees still scored lower than TBL attendees.

It is striking that student performance in the iRAT was moderate to low (around 40%), implying that students have below-average initial knowledge. The higher difficulty index and the poor performance of students in the iRAT could be explained by the fact that the iRAT was a formative assessment. It is known that formative exams often result in low scores as passing these exams is not obligatory, a phenomenon also observed in our students [24]. Despite this, the iRAT signals possible knowledge gaps and stimulates problem-solving skills. In line with this, it is important to mention that the placement of the TBL session prior to the exam could have resulted in an added advantage of revisiting and revising topics before the final exam.

No statistically significant correlation was found between the iRAT scores and the pharmacology-related questions on the final exam. The considerable disparity in performance between TBL attendees and non-TBL attendees highlights the value of TBL in facilitating knowledge application and problem-solving skills essential for success in pharmacology examinations. Nevertheless, it is noteworthy to highlight that not all students scored higher on the pharmacology-related final exam question than on the iRAT. Given that the iRAT score did not directly impact the academic outcome of the student, one plausible explanation for the diminished performance on the final exam could be attributed to examination-related stress leading to suboptimal academic performance [25].

### 4.1. Implications of Results for Pharmacology Educators

TBL as an active learning methodology: This work aims to encourage educators to implement TBL as an engaging and effective teaching strategy that could be implemented for teaching pharmacology. The findings in this study support the substantial value of TBL as an effective means of stimulating active learning in pharmacology education. This aligns with earlier studies in other medical contexts in which TBL was used to teach pharmacology, providing additional evidence of its effectiveness [26,27,28]. TBL sessions can serve two purposes: firstly, they provide students with a ‘reality check’ or immediate feedback on their level of knowledge and the depth one requires to solve presented clinical problems. This is also reflected in the evaluation of the TBL session, where students indicated a low level of confidence on the subject matter, possibly stimulating them to study more for the exam. Secondly, discussing the iRAT questions and clinical problems during the TBL session promotes deeper understanding of the subject material, preparing students for solving complex clinical problems. Students who did not attend the TBL session missed this opportunity. Although these students had access to the materials, they did miss the interaction and the in-depth discussion, both crucial elements of in-depth learning.

Placement of TBL sessions within the curriculum: TBL, as it incorporates elements such as formative testing, feedback and solving authentic clinical problems, could be an ideal educational tool for revising topics prior to exams. These elements of TBL could help students to discover their knowledge deficits and to optimally prepare themselves to solve clinical cases. This could be especially helpful in institutions where frequent and standard TBL sessions cannot be arranged due to varying restrictions.

Modifications in TBL approach might be essential to fit educational different contexts. Modifications in TBL needed to fit the educational context should be evidence-driven. Several elements of TBL have been considered important, such as carefully formed and managed teams, timely feedback, problem solving, and a competitive element [20]. However, due to limitations in organizational structure, availability of learning spaces, and financial constraints, modifications in the TBL sessions may be needed to fit the educational setting. In the current setting, students had full autonomy in forming groups, and the competitive element of TBL was not incorporated. The rationale behind this is based on the self-determination theory. Students were allowed to choose their groups to promote a sense of autonomy, and a sense of relatedness was promoted by creating a safe (non-competitive) space where students could practice and make mistakes [29]. Promoting competitiveness could negatively affect student motivation [29] but also negatively impacts equity [30]. The decision to deviate from the classic TBL structure also has a pragmatic reason, as none of the students of the course were obliged to attend all classes. Therefore, prior formation of the groups was not convenient as students might not attend the TBL session. Additionally, it would also lead to extra administrative work for the teachers. Our results demonstrate that despite modifying these elements of TBL, it still had a positive impact on student performance.

The current TBL session, including the iRAT and final exam, was conducted and documented digitally using various online teaching platforms. However, TBL sessions could also be conducted without digital teaching platforms. Elements of TBL such as promoting peer discussion [31], solving authentic problems, activating knowledge through questions, and formative assessments [32] are known to positively affect learning, even outside the TBL context. This could explain the positive effects of TBL in our cohort. Adjusting TBL to fit the educational context can make integration of the TBL method more feasible. However, such adjustments should be guided by educational scholarship, especially prior to large-scale roll-out. Based on our experience, using a variety of active teaching methodologies, TBL being one of them, usually helps to break the monotony for students and teachers.

### 4.2. Future Directions

Further study is required to investigate the impact of integrating TBL on students’ learning strategies. It is essential to evaluate the immediate learning outcomes but also the long-term approach to learning adopted by students after the introduction of TBL. Furthermore, additional research should be directed to extending the influence of TBL on long-term knowledge retention and academic performance. This could give insights into sustained knowledge retention over time and thereby the enduring impact of TBL. Furthermore, it would be interesting to interview students on the benefits of TBL compared to passive teaching methods. Unfortunately, we could not do this, as students were already overwhelmed by the sheer number of evaluations (survey fatigue) [33] they were exposed to and we had to choose low-threshold, to-the-point questions. Increasing the length of the questionnaire could have led to fewer responses [23].

### 4.3. Strengths and Limitations of the Study

Strengths: Herein, we show the importance of adjusting established teaching methodologies by identifying and including the crucial underlying educational principles to fit into individual curricula. Often, excellent teaching methodologies cannot be implemented due to rigid frameworks and logistic reasons. Our study shows that by understanding the underlying educational principles (in this case formative testing, feedback, and problem-solving skills), educators can introduce innovative teaching methods into their curricula with positive outcomes. The barriers to incorporating TBL could be additional workload for teachers, i.e., converting existing lessons into a TBL format, logistic reasons, and student non-acceptance of the teaching method. Especially for the latter, initial resistance could be expected if students are exposed to traditional teaching methods requiring minimal interaction with teachers or students. Similarly, students might not be open to the prior preparation that TBL requires. It is known that students feel that they have a better grasp of the subject matter when following passive learning methods than when exposed to active learning [5]. Unsurprisingly, however, the performance of students participating in active learning is better. If students understand the benefits of TBL, participation and involvement could be improved, leading to better performance. In this light, adapting teaching methodologies to fit existing curricula is needed. Initially, this could be experienced as time-consuming. Nevertheless, it will benefit students in the long run.

Limitations: It is important to acknowledge certain limitations of this study. The current study was conducted in a naturalistic setting, meaning the TBL teaching form was implemented within the existing curriculum. Although this is a strength on the one hand, as our findings are of immediate relevance for our educational setting, the results are limited by a non-randomized sample of learners. This raises the possibility of existing differences between both groups. For example, TBL students may have been more motivated or may have been less confident in their pharmacology knowledge compared to non-attendees. However, randomization was not possible in our setting due to the nature of lessons being non-obligatory. For this reason, the study setting was kept as natural as possible, resulting in groups based on their attendance of the TBL session. Second, academic confidence can be influenced by experience but also by societal factors within or around the academic setting [34]. This study did not correct for possible differences in individual student characteristics. Third, self-assessed perceptions of knowledge, often used as a tool to guide student learning, can be influenced by various individual factors, possibly resulting in self-reporting bias. However, within medical education, self-assessment is considered a credible method for educational program evaluations [35]. Finally, while the study design allowed for comparisons between TBL attendees and non-attendees, factors such as prior pharmacological knowledge and individual study habits may have influenced outcomes. The results must be viewed with these limitations in mind.

## 5. Conclusions

The current findings identify TBL as an engaging and effective instructional approach for teaching pharmacology, promoting active learning and practical application of knowledge. The current study emphasizes the importance of adapting teaching methodologies to suit a given educational context, with modifications guided by an evidence-based approach. By integrating TBL’s interactive and collaborative elements, educators can enhance student engagement and deepen their understanding of pharmacological concepts. Consequently, TBL could be incorporated into the existing pharmacology teaching method, promising enriched learning experiences and improved academic outcomes. This aligns with the goal of active learning and equipping students with the skills necessary for safe and efficient medication prescribing.

## Figures and Tables

**Figure 1 pharmacy-12-00091-f001:**
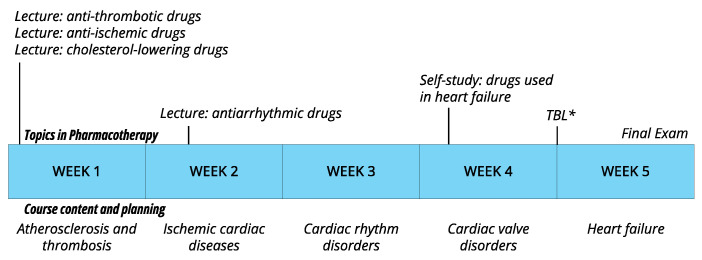
Schematic representation of the 5-week course on cardiovascular diseases during which cardiovascular pharmacology was taught. The content, topics in pharmacotherapy, and planning are indicated per week. Educational activities and materials were accessible to all students. * TBL = the TBL session, which was attended by a part of the cohort, based on which the two groups (TBL attendees and the non-TBL attendees) were formed.

**Figure 2 pharmacy-12-00091-f002:**
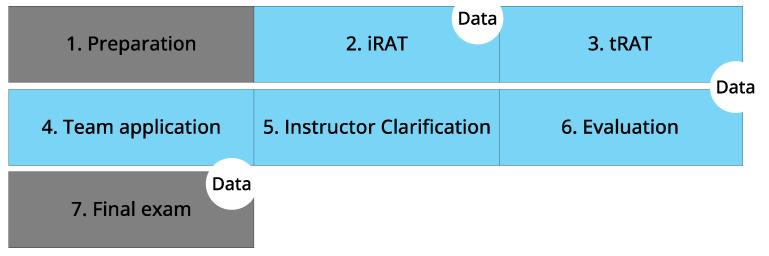
Schematic representation of the various components of the study including the elements of TBL. The grey boxes indicate teaching elements not a part of the TBL session. The blue boxes indicate the steps undertaken during TBL session. Time points for data collection are indicated in white circles. iRAT = individual readiness assurance test; tRAT = team readiness assurance test.

**Figure 3 pharmacy-12-00091-f003:**
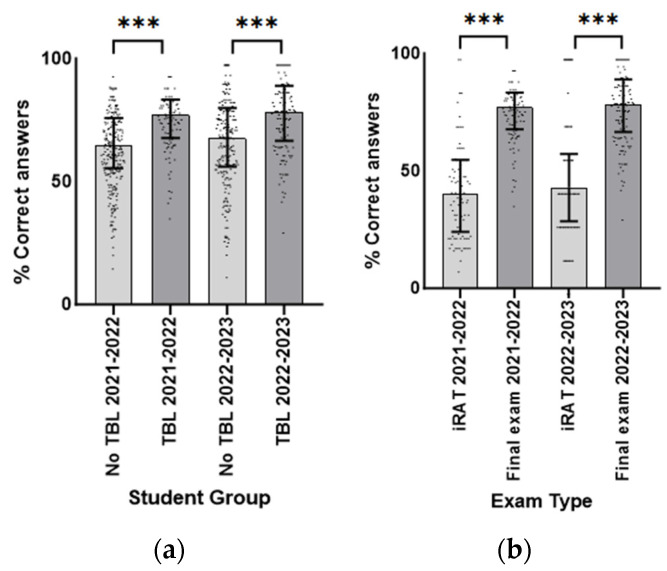
(**a**) Student performance in final exam for TBL attendees and non-TBL attendees in 2021–2022 and 2022–2023. Values are median (IQR). *** = *p* < 0.001. (**b**) Student performance in iRAT and final exam for TBL attendees in 2021–2022 and 2022–2023. Values are median (IQR). *** = *p* < 0.001.

**Table 1 pharmacy-12-00091-t001:** Student perceptions of team-based learning. Frequencies for each Likert scale option, mean (ME), and median (MD), are provided for each statement. Likert scale key: SD: strongly disagree; D: disagree; N: neutral; A: agree; SA: strongly agree.

		SD (1)	D (2)	N (3)	A (4)	SA (5)	ME	MD
1	The TBL session has aided in achieving the learning objectives.	0	4	15	72	52	4.2	4
2	The structure of the TBL session was good.	0	3	19	62	59	4.2	4
3	The organization (planning, supervision, etc.) of the TBL session was good.	0	0	16	60	67	4.4	4
4	The content of the TBL session was informative.	0	2	4	61	76	4.5	5
5	I think I have sufficient knowledge on this subject.	7	26	54	46	10	3.2	3
6	I learnt a lot from the collaboration during the TBL session.	1	14	42	59	27	3.7	4
7	I learnt a lot from the practice questions.	0	3	18	51	71	4.3	4
8	The patient’s story has contributed to learning about the application of medicines in real life.	2	15	45	57	25	3.6	4

## Data Availability

The research data are not publicly available, but could be available from the corresponding author on reasonable request, provided prior permission is granted by UMC Utrecht.

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
