# Peer review of "Using Team-Based Learning to Teach Pharmacology within the Medical Curriculum"

_pharmacy, 2024, doi:10.3390/pharmacy12030091_

Round 1
Reviewer 1 Report
Comments and Suggestions for Authors
I read with interest the paper titled "Using Team-Based Learning to Teach Pharmacology within the Medical Curriculum"
Minor suggestion:
1. References that appear [1],[2] should appear as [1,2]
2. Figure 1 provides a detailed structure of the course [Figure 1]. - no need to add 2 times the information "Figure 1"
3. Have you recorded any chareacteristics of the students that could be compared (eg: age, number of times repeating course, number of the hours following TBL-sessions? Those characteristics are from high interest in the discussion.
4. Have authors tested the normality of data? Why the use of non-parametric tests?
5. Table 1 formatation is very hard for readers. Please format it better.
6. Figure 3, graph b) - Can authors please provide the plot (and statistical diferences calculation) to the same parameters (iRAT and final examination) to no TBL students.
Author Response
Please find the comments attached

Reviewer 2 Report
Comments and Suggestions for Authors
Overall, a clearly written manuscript with detailed description of the methodology. However, the main limitation of this study is the design. While it is meant to be naturalistic, the TBL group had an additional learning session compared to the non-TBL group. The comparison for the final scores between the TBL and non-TBL group is therefore unfair as the TBL group had an additional learning session where they were able to practice through quiz and case discussions. This already puts this group of students at an advantage versus the non-TBL group which appear to not have received any practice questions.
It was also challenging to visualize the interventions that a student underwent due to many components being voluntary in terms of attendance. It was also unclear if the TBL session focused only on the topic based on preparatory study assignment on heart failure or all topics which included lecture topics on antithrombotics, antiischemic, anticholesterol and antiarrhythmic drugs. It was difficult to interpret the perception survey results on whether students viewed the TBL method as the in-person session only, or if it was viewed inclusive of the pre-class interventions which included lectures and pre-readings.
The current manuscript has its merits but perhaps keep that merits to the TBL group and not compare performance with the non-TBL group.
Specific comments are as follows:
1) Introduction: The authors should briefly outline the structure of TBL in the introduction e.g. it involves individual readiness assurance test, followed by team readiness assurance test and subsequent application case especially since they adopted a very specific approach of TBL by Prof Michaelson.
2) Methods: Could the authors clarify that the TBL group essentially had an additional intervention which is the TBL session as compared to the non-TBL group? In other words, both groups received the same number of interventions e.g. lecture, interactive lecture, small group teaching, but the TBL group had an additional session which was the TBL session?
3) Methods: Figure 1 was helpful but perhaps you want to re-draw it to show what the 2 group of students underwent i.e. which parts were similar, and which parts diverged?
4) Methods: For the TBL group, what did the “preparatory study assignment” entail? Was it readings or pre-recorded lectures? For the non-TBL group, how did they learn the topic of heart failure medications?
5) Methods: Could the authors clarify more about the quizzes ran during the TBL session:
a. Were they testing all topics, not just heart failure medication?
b. How many questions were there?
c. Which level of Bloom’s taxonomy did these questions test?
d. For the multiple correct answer questions in 2021-22 and final exam, were these marked as all-or-none or partial score was given if students selected some correct answers?
e. Were the questions validated by another content expert?
f. Were the rigour/level of difficulty for both cohorts matched?
g. Were the rigour/level of difficulty between the iRAT/tRAT quizzes similar to final exam?
h. The final exam comprised 8 MCQ with 1 being multiple answer question. What is the distribution of type of questions amongst these e.g. recall, application, analysis etc? This is important as the authors stated on Pg 7, Section 3.2.3 that “…TBL in facilitating knowledge application and problem solving skills…. “
6) For the 8-statements on perception of students towards TBL:
a. Is this specific to the in-person session?
b. Statement 5 “I think I have sufficient knowledge on this subject” pertains to which topic? Again, is it in relation to lectures and prep session on heart failure topic, further reinforced with quizzes and discussion during TBL?
c. Statement 7 “practice questions” refers to the iRAT and tRAT?
7) For the reported median scores, are these in percentages?
8) The median score for the iRAT is very low at only about 40(%). This is close to failing. What was the reason fr this?
9) One of the important value of TBL is peer-learning. tRAT scores often improved from iRAT. Did the authors conduct a comparison between the iRAT and tRAT scores? It is important to present this as evidence which further supports the value of peer learning in TBL. This is especially since the median score for iRAT is very low at 40+%
10) Section 3.2.3: I am not very sure about the value of making this correlation between iRAT score and final exams, since the non-TBL group did not have a chance to attempt any practice quiz questions.
Author Response
Please find our comments attached.

Reviewer 3 Report
Comments and Suggestions for Authors
Some aspects relating to the study design and methods could be clarified to the reader. For example:
-Only 1 TBL session was administered, in an interactive mode, with max 48 participants. what was instructor-to-student ratio? as in how many intructors were needed to manage a team of 48 participants and if this workload is reasonable ?
-Was one class/TBL sufficient to evaluate the effectiveness of the approach for the whole cardiovascular curriculum?
-how long was a session? was the time always enough to complete all the components of the session?
-students who refrained form the session, what were their educational components? only lectures over the 5 weeks then the exams?
-What were the strengths and weekness of the approach?
-The authors describe a modified TBL, compared to what, please describe the standard TBL breifly in the Introduction.
-As TBL session was adminstered just before the exams,
may be atttendees had the advantage of revising their knowledge before the exams? could thus be a factor for the better scores?
-Figure 1 has several English typing mistakes.
section 2.3 Procedure:
-was "Step 1. Preparation" not available for non attendees? please clarify.
-which components were part of the active session ?
may be the authors should break down the items into sections to distinguish between the prior activities and the active session activities
e,g,
1. preparation
2. in-class active learning (please elborate on the activities, their timing and composition as the students start the session)
3. feedback
-figure 2: please indicate the color difference. what is the standard teaching methods that the non-attendees were exposed to?
-Table 1. no open ended questions or free text comments by the students in the questionnaire? this could have enriched the feedback given to aspects not covered by the questionnaire. could this be a recommendation for future score assessment?
-Discussion:
Please cite other work or a sytematic review that show similar/discrepant results in addition to the 2 studies mentioned.
-what are the facilitators and barriers to implememt this approach as a part of the standard eductional process at other universities?what were the most beneficial features of the approach and the most difficult to manage that still needs improvement? which tools are necessary to perform the activities and the evaluation, being regarded as "minimum requirements" e.g. Computers or tablets, reliable internet access, educational software or platforms,
-should the session be repeated , i.e. on more weeks?
-how many credit hours for TBL vs regular teaching did the students recieve?
Author Response
Please find our comments attached.

Round 2
Reviewer 2 Report
Comments and Suggestions for Authors
I appreciate that the authors have made significant edits to clarify the methodologies further and even conducting further analysis e.g. the rigour of the assessments. I have no further comments. The paper was clearly written and would add another body of evidence to support the benefits of TBL in education.
Reviewer 3 Report
Comments and Suggestions for Authors
Dear Authors,
Thanks for your responses which adequatley addressed all th comments raised.